# Critical Barriers to Industry 4.0 Adoption in Manufacturing Organizations and Their Mitigation Strategies

Ahmed Sayem [1], Pronob Kumar Biswas [1], Mohammad Muhshin Aziz Khan [1], Luca Romoli [2,*] and Michela Dalle Mura [2]

1   Department of Industrial and Production Engineering, Shahjalal University of Science and Technology, Sylhet 3114, Bangladesh
2   Department of Civil and Industrial Engineering, University of Pisa, Largo Lazzarino, 1-56122 Pisa, Italy
*   Correspondence: luca.romoli@unipi.it

**Abstract:** The fourth industrial revolution, fueled by automation and digital technology advancements, enables us to manage manufacturing systems effectively. Its deployment in enterprises has now become increasingly important in developed and emerging economies. Many experts believe that barriers associated with Industry 4.0 implementation are critical to its success. Therefore, this study aimed to identify the major hurdles to Industry 4.0 adoption and reveal their interrelationships. Initially, the literature was thoroughly studied to determine the sixteen barriers impeding I4.0 adoption. Then, based on experts' opinions, an integrated fuzzy-DEMATEL approach was utilized to examine the most significant challenges to I4.0 deployment. The results demonstrated the distribution of barriers in which the economic dimension played a decisive role, affecting technological, regulatory, and organizational dimensions. As observed in the barrier mapping, the lack of qualified workforce was a typical adoption barrier. Finally, the mitigation strategies developed would help managers to overcome the identified critical obstacles.

**Keywords:** Industry 4.0; barriers; fuzzy-DEMATEL; manufacturing; developing economy





## 1. Introduction

Nowadays, global competition and the necessity for quick adaptability to changing market demands drive industrial production. Competition push and technology pull are indeed two of the forces that make manufacturing companies move in today's competitive business environment. On the one hand, there is an enormous competition push characterized by a growing need for product-development time reduction, greater demand for mass personalization, higher requirements for manufacturing flexibility, real-time decision making, and maximization of resource utilization [1,2]. Mass customization at a low cost and with a short lead time is desired and has evolved into a differentiating as well as a survival factor [3]. On the other hand, there is a pull from technological advancement, which is characterized by enhanced process automation and digitalization innovations, the development of trouble-free semantic integration, progress in networking technologies, cost reduction, increasingly rapid obsolescence of products and ongoing miniaturization of electronic devices [4,5]. Because of the changes that manufacturing organizations have faced, they have devoted significant effort to developing and implementing smart technology in their production system [6]. Thereafter, administered by these two-way driving forces and tempted by future expectations, the term 'Industry 4.0' was established ex-ante for a planned advancement, titled 'Fourth Industrial Revolution.'

The world has already gone through three industrial revolutions, each of which has profoundly altered our modern society. Each of these three developments—mechanization (Industry 1.0), mass production (Industry 2.0), and automation (Industry 3.0)—resulted in significant changes in manufacturing and production activities. Recently, in the history

of industrial development, academics and industry professionals have identified a new transformation, referred to as Industry 4.0. Industry 4.0′ was officially coined at the 2011 Hannover Fair in Germany [7]. It has commonly been assumed that, in the not-too-distant future, such an industrial revolution is inevitable [8]. Noteworthy to mention that, for the first time, an industrial revolution is predicted ex-ante, not observed ex-post [9]. Thus, countries, research institutes, and companies are working relentlessly to shape the future proactively for this planned industrial revolution. For the moment, emerging technologies and innovations, viz., the internet of things, artificial intelligence, autonomous robots, big data, cloud computing, simulation/digital twin, and augmented reality, are diffusing much faster and more widely than in previous ones [10]. Furthermore, increasing uncertainties in today's global market have forced manufacturing industries to introduce technological innovations to improve profitability and competitiveness. Recently a plethora of research suggests that the technologies linked with Industry 4.0 have the potential to enhance operational efficiency, improve responsiveness, boost traceability, strengthen capacity utilization, reduce costs, but also enhance ergonomics of workers, as also promoted by 2030 Agenda for Sustainable Development [11–15]. Consequently, the Industry 4.0 phenomenon has gained paramount importance, and its adoption by manufacturing enterprises has become a top priority for developed and developing economies. In summary, Industry 4.0 swiftly became a catchphrase and a blooming research issue in academia and the industrial sector due to its multifaceted importance and viewpoints.

In the new global economy, the adoption of Industry 4.0 has emerged as a central issue for manufacturing firms. Accordingly, to reap the full benefits of this new industrial revolution, countries and organizations are taking the necessary initiatives to implement it. For example, the German federal government has made Industry 4.0 a key component of its high-tech strategy to ensure the competitiveness of the German economy [16], and China has developed the "Chinese Manufacturing 2025" plan to accelerate manufacturing's transition from Industry 3.0 to Industry 4.0 to become a global leader [17]. To embrace the benefits of the fourth industrial revolution, similar competitive strategies were launched in other geographies under different names, for instance, the United States' Advanced Manufacturing Partnership [18], Japan's Society 5.0 [19], South Korea's Manufacture Innovation 3.0 [20], and Sweden's Production 2030 [21]. In addition, data from several studies tried to predict that the economic impact of Industry 4.0 would be vast and sustainable, as it notably improves operational effectiveness and supply chain competencies [12,22]. It is commonly accepted that the Industry 4.0 adoption process is technically challenging and cannot be managed if the effect of barriers is neglected [22–25].

Despite the significance and numerous benefits highlighted in published articles and reports, recent research suggests that manufacturing organizations face challenges in making the transition to Industry 4.0 from the previous stage [26]. Thus, the futuristic idea of Industry 4.0 is still only a vision for most manufacturing companies, especially those in a developing economy [27]. Since identifying adoption barriers and their interrelationships provides insights for developing efficient and effective strategies to accelerate Industry 4.0 implementation, analyzing the adoption barriers has been deemed crucial. Thus, it is necessary to understand the barriers and their interrelationships to lower the risk of failure and encourage early adoption of Industry 4.0. Recognizing such importance, there is a growing corpus of research targeted at identifying barriers to Industry 4.0 adoption in various geographic areas and industrial sectors. Cugno et al. [28], for example, identified hurdles to Industry 4.0 adoption in Italy through literature reviews, in-depth interviews, and multiple case studies. Based on interviews with top executives of leading manufacturing enterprises in Hungary, Horváth and Szabó [29] identified impediments to adopting Industry 4.0. Through feedback from industry and academic experts, Kamble et al. [30] examined barriers to Industry 4.0 adoption in manufacturing organizations and offered a priority ranking of the barriers. Through a focus group study, Orzes et al. [31] empirically determined the key impediments to Industry 4.0 implementation in small and medium-sized enterprises (SMEs). Kiel et al. [32] examined the barriers from social, environmental, and financial

perspectives. Notably, identified barriers to Industry 4.0 adoption differ by country [25], as the obstacles are not only technological but also organizational and socio-economic in nature [33]. Moreover, a disparity exists between developed and developing countries regarding knowledge, readiness, and adoption of Industry 4.0 [34]. Nonetheless, very little is known about the nature of factors at the moment, and the causal factors leading to the emergence of Industry 4.0 remain speculative. Furthermore, few studies have compared the barriers crucial for two different economies or geographical locations.

Therefore, this paper seeks to fill a gap in the literature and helps decision makers formulate strategies to mitigate the critical barriers to adopting Industry 4.0. This research aims to answer the following questions:

1.  Considering the perceived barriers to adopting Industry 4.0, what is the causal relationship among the critical barriers?
2.  How are Industry 4.0 adoption barriers related to various economic orientations as well as geographical locations?
3.  How can manufacturing enterprises willing to adopt Industry 4.0 overcome the identified critical barriers?

The manuscript is organized as follows: Section 2 provides a literature overview on Industry 4.0 adoption barriers and tools and techniques for analyzing such obstacles. Section 3 describes the methodology used for this study. This section also presents the approach employed to identify the critical barriers and develop the causal inter-relationships diagram. Section 4 reports the main results obtained through quantitative analysis. Section 5 first presents the comparative analysis of barriers from different economic perspectives. Later, it looks at mitigation strategies required to overcome the critical barriers. Finally, Section 6 concludes the study by highlighting the principal contributions and implications and proposing avenues for future research.

## 2. Literature Review

There are considerable differences between Industry 4.0 and the prior three industrial revolutions in terms of their scope, size, and complexity. While the previous three industrial revolutions arose to perform muscle work for humankind, Industry 4.0 emerged to perform brain functions as well [35]. Industry 4.0 represents an entirely novel form of a production system, enabling wholly automated and human-independent machines and self-managing processes in which the devices can communicate with each other [36,37]. Current widespread practiced mass production is anticipated to convert into mass customization with increased production speed, higher product quality, and optimized efficiency through data-driven decision-making [33,38]. However, the vast and rapidly growing literature has demonstrated that companies across the globe face various barriers to adopting Industry 4.0 into practice [22,25]. The following paragraphs will discuss a more detailed account of the relevant literature.

Chauhan et al. [39] identified twenty challenges to Industry 4.0 adoption in Indian manufacturing firms, categorizing them as intrinsic (related to a firm's internal environment) and extrinsic (related to a firm's external environment). In addition, they used Structural Equation Modeling (SEM) to examine the consequences of these barriers, most notably on supply chain competency and operational performance. Ultimately they concluded that both intrinsic and extrinsic barriers hamper the adoption of Industry 4.0 and that overcoming these obstacles can considerably improve the firm's operational performance and supply chain competency. Stentoft et al. [40] used a mixed-method (qualitative and quantitative) approach to investigate the barriers to Industry 4.0 practice among Danish small and medium-sized manufacturers. A lack of both human and financial resources, a lack of understanding of how technology interacts with humans, a lack of understanding of the strategic importance of new digital technologies, and the employees' need for continuous training were the most significant factors identified in their study. Raj et al. [41] used the Grey Decision-Making Trial and Evaluation Laboratory (DEMATEL) approach to examine the hurdles to deploying Industry 4.0 technologies in the Indian manufacturing

sector. After conducting a thorough literature review, they identified fifteen barriers and arranged them in a hierarchical order using feedback from six industry experts. They found (i) a lack of standards, regulations, and forms of certification, (ii) a lack of internal digital training, and (iii) a lack of infrastructure as the top three critical obstacles to adopting Industry 4.0. Veile et al. [42] explored the challenges to Industry 4.0 implementation in German industries based on semi-structured interviews with experts. The top three barriers identified in their study were technical integration, organizational transformation, and data security. Obiso et al. [43], via a comprehensive literature review followed by a focus group meeting, identified twenty-four adoption barriers of Industry 4.0 for manufacturing industries. They grouped these challenges into three categories based on their impact on Industry 4.0 adoption: technological, economic and regulatory, and social. Technological hurdles were the most prevalent of these three categories, accounting for twelve out of twenty-four barriers. The low maturity level of technologies, lack of integration, data inconsistency, and weak stability were some of the technological barriers highlighted in this study. Karadayi-Usta [44] detected nine impediments to Industry 4.0 adoption in Turkish manufacturing organizations and investigated their relationships using Interpretive Structural Modeling (ISM) analysis based on expert opinion. According to this analysis, the key hurdle to Industry 4.0 adoption is an ineffective education system. From 176 SME managers' responses collected through a questionnaire survey in Romania, Türkeş et al. [45] highlighted multiple challenges enterprises might face in implementing Industry 4.0 technologies. The crucial barriers found in this study were a lack of understanding of Industry 4.0, a lack of human resources, and a lack of standards.

Using the Best-Worst Method (BWM), Moktadir et al. [46] identified, evaluated, and ranked the barriers to implementing Industry 4.0 in the context of process safety and environmental protection of the leather industry. They ranked the obstacles according to the opinions of eight industry experts and identified the lack of technological infrastructure, the difficulty of reconfiguring production patterns, and data security as the top three challenges. Glass et al. [47] selected fifteen barriers from the literature and tested hypotheses using data obtained via online surveys of 176 participants from German industries. Through hypothesis testing, they revealed that only seven of these barriers obstruct Industry 4.0 deployment. In descending order of importance, these statistically valid barriers were: missing standards, risk of data loss and external manipulation, low maturity level of new technologies, missing skilled workers and know-how, poor external conditions (infrastructure, legal issues), missing cooperation partners and funding programs, and difficulty in formulating an Industry 4.0 strategy.

Turning now to the evidence from the literature on other industrial sectors. Da-Silveira et al. [48] conducted a systematic review of the literature to identify the barriers to Industry 4.0 adoption in the agriculture industry across five dimensions: technological, economic, political, social, and environmental. In their report, they came up with twenty-five barriers, and they proposed that several of them should be dealt with first, including component incompatibility, reliability concerns, infrastructure inadequacies, requirements to foster R&D and innovative business models, a lack of digital skills and/or skilled labor, information asymmetry, and educational issues. Alaloul et al. [49] investigated the difficulties associated with adopting Industry 4.0 technology in the construction industry. They collected data using a questionnaire and ranked the challenges encountered using the Relative Importance Index (RII). This study also discovered that social and technological issues were major hurdles to successful implementation. Ajmera and Jain [50] conducted an extensive literature review and gathered opinions from industry and academic experts to identify barriers to Industry 4.0 adoption in the healthcare industry. They reported top management support, exclusive and skilled workforce requirements, inadequate maintenance support systems, and political support as major obstacles using the Total Interpretive Structural Modeling (TISM) approach.

The above literature review makes it clear that the research agenda of identifying and understanding hurdles to Industry 4.0 adoption is critical for overcoming them and

paving the way for the successful deployment of Industry 4.0 in manufacturing and other sectors. To ascertain and better understand such hurdles, most researchers collected data from experts using semi-structured interviews, focus group meetings, as well as online and offline questionnaire surveys. Statistical tools and multi-criteria decision-making (MCDM) techniques were used extensively in the studies. Until now, the best-worst method (BWM), interpretive structural modeling (ISM), total interpretive structural modeling (TISM), structural equation modeling (SEM), and Grey Decision-Making Trial and Evaluation Laboratory (DEMATEL) have all been frequently employed. Table 1 delineates sixteen recognized barriers to the adoption of Industry 4.0 along with respective sources.

**Table 1.** Industry 4.0 adoption barriers.

| Barrier' Dimensions | Code | Industry 4.0 Adoption Barriers | Source |
|---|---|---|---|
| Technological | C1 | Technology Availability and Compatibility | Cugno et al. [28], Horváth and Szabó [29], Obiso et al. [43], Aggarwal et al. [51] |
| | C2 | Low Maturity of Technology and Seamless Integration | Kamble et al. [30], Obiso et al. [43], Bakhtari et al. [35], Jain and Ajmera [52] |
| | C3 | Information Technology Infrastructure | Kamble et al. [30], Moktadir et al. [46], Glass et al. [47], Kumar et al. [53] |
| | C4 | Cyber-Security and Privacy | Chauhan et al. [39], Raj et al. [41], Ghobakhloo [54], Sony and Naik [55] |
| | C5 | Capability to Manage Big Data | Raj et al. [41], Obiso et al. [43], Kumar et al. [54], Genest and Gamache [56] |
| Economical | C6 | Requirement for High Initial Investment | Cugno et al. [28], Horváth and Szabó [29], Kamble et al. [30], Da-Silva et al. [57] |
| | C7 | Uncertainty of Return-On-Investment | Horváth and Szabó [29], Kamble et al. [30], Bakhtari et al. [35], Kumar et al. [53] |
| Regulatory | C8 | Availability of Reference Architecture and Standards | Obiso et al. [43], Türkeş et al. [45], Glass et al. [47], Trappey et al. [58] |
| | C9 | Government Support and Legal Issues | Cugno et al. [28], Kamble et al. [30], Glass et al. [47], Aggarwal et al. [51] |
| | C10 | Complexity in Supply Chain Integration and Coordination | Horváth and Szabó [29], Raj et al. [41], Obiso et al. [43], Kumar et al. [53] |
| | C11 | Employee Fear and Resistance to Change | Chauhan et al. [39], Raj et al. [41], Moktadir et al. [46], Kumar et al. [53] |
| Organizational | C12 | Education and Training Programs | Horváth and Szabó [29], Türkeş et al. [45], Bakhtari et al. [35], Masood and Sonntag [59] |
| | C13 | Knowledge, Awareness, and Competence of Industry 4.0 | Horváth and Szabó [29], Moktadir et al. [46], Türkeş et al. [45], Narwane et al. [60] |
| | C14 | Management Commitment and Leadership | Yüksel [33], Bakhtari et al. [52], Huang et al. [61], Kumar et al. [62] |
| | C15 | Availability of Skilled Workforce | Horváth and Szabó [29], Glass et al. [47], Kumar et al. [53], Hamzeh et al. [63], |
| | C16 | Organization Structure and Culture | Cugno et al. [28], Horváth and Szabó [29], Bakhtari et al. [35], Narwane et al. [60] |

Based on the aforementioned considerations, the present study aims to identify the major hurdles to Industry 4.0 adoption, making use of a comprehensive literature review as well as discussions with industry experts, and explain their causal relationship. To this purpose, an integrated Fuzzy Decision-Making Trial and Evaluation Laboratory (DEMATEL) approaches is used.

## 3. Methodology

One of the primary concerns of this study is to explain the causal relationships between barriers to Industry 4.0 adoption in manufacturing industries. Due to (i) the interdependency and complex nature of the barriers, as well as (ii) the uncertainty and imprecise nature of expert evaluations, traditional models are inadequate in achieving this goal. Fuzzy set theory can be used to deal with fuzzy and imprecise data, and in most cases, intuitive fuzzy sets from four to six experts are enough to reach a reliable conclusion [64]. In addition, DEMATEL (Decision Making Trial and Evaluation Laboratory) can evaluate complex systems with causal relationships between contributing factors based on expert opinions. More importantly, results obtained from the analysis of such methods provide valuable insight into the relative importance of each factor to the system as a whole [65]. Therefore, an integrated approach of combining fuzzy set theory and the DEMATEL can produce a more concrete conclusion than traditional approaches and thus widely used

in many areas, practically in computer science, engineering, business and management, decision sciences, and social sciences [66–69]. Therefore, a combination of fuzzy set theory and the DEMATEL methods was used in the data analysis. The following sub-section discusses both methodologies.

*3.1. Fuzzy Set*

The fuzzy set, introduced by Zadeh in 1965, is a powerful method for overcoming the vagueness, inconsistency, and uncertainty of human judgment and assessment in decision making [70]. Fuzzy sets are commonly used in real-world problems where the environment is uncertain, indeterminate, and ambiguous. In fuzzy set theory, linguistic variables are transformed into fuzzy numbers according to membership functions, and those fuzzy numbers are defuzzified to achieve crisp scores [71]. Each number between 0 and 1 in a fuzzy set is a partial true value, corresponding to Boolean logic: 0 or 1, and is denoted through membership functions. Triangular membership function, trapezoidal membership function, Gaussian membership function, and sigmoid membership function are examples of membership functions. Among those, a common and widely used membership function is the triangular membership function, which is represented as a triplet A $^\sim$ = (a_1, a_2, a_3). These triplets (a_1, a_2, a_3) are real numbers that represent the lower, medium, and upper numbers of the fuzzy sets (a_1 ≤ a_2 ≤ a_3). Based on the remark put by Zhao and Bose [72]—the triangular membership function is superior to all other membership functions-, this membership function was chosen for this study. The relationship between linguistic terms and triangular fuzzy numbers is presented in Table 2.

**Table 2.** Relationship between linguistic terms and triangular fuzzy numbers.

| Linguistic Terms | Triangular Fuzzy Numbers |
| :---: | :---: |
| No influence (No) | (0.00, 0.00, 0.25) |
| Very low influence (VL) | (0.00, 0.25, 0.50) |
| Low influence (L) | (0.25, 0.50, 0.75) |
| High influence (H) | (0.50, 0.75, 1.00) |
| Very high influence (VH) | (0.75, 1.00, 1.00) |

Defuzzification is the conversion of fuzzy numbers into specific scores, also referred to as crisp values. It is necessary to perform defuzzification for further aggregation. This study employs the CFCS (Converting Fuzzy data into Crisp Scores) defuzzification method to obtain specific values. The CFCS method, proposed by Opricovic and Tzeng [73], is based on determining the left and right scores through fuzzy minimum and fuzzy maximum, and the total score is determined as a weighted average based on the membership functions. The CFCS method consists of the following five steps:

Let, $\widetilde{\omega}_{ij}^{k} = \left( a_{1ij}^{k} , a_{2ij}^{k} , a_{3ij}^{k} \right)$; indicate the fuzzy assessments, the degree of criterion $i$ that affects criterion $j$, of evaluator $k$ ($k$ = 1, 2, 3, . . . , $k$).

Step 1: Normalize the Triangular Fuzzy Numbers

$$\begin{aligned}
xa_{1ij}^{k} &= \frac{a_{1ij}^{k} - \min a_{1ij}^{k}}{\max r_{ij}^{n} - \min l_{ij}^{n}} \\
xa_{2ij}^{k} &= \frac{a_{2ij}^{k} - \min a_{2ij}^{k}}{\max r_{ij}^{n} - \min l_{ij}^{n}} \\
xa_{3ij}^{k} &= \frac{a_{3ij}^{k} - \min a_{3ij}^{k}}{\max r_{ij}^{n} - \min l_{ij}^{n}}
\end{aligned} \tag{1}$$

Step 2: Compute Right (*rs*) and Left (*ls*) Normalized Values

$$\begin{aligned}
xrs_{ij}^{k} &= \frac{xa_{3ij}^{k}}{1 + xa_{3ij}^{k} - xa_{2ij}^{k}} \\
xls_{ij}^{k} &= \frac{xa_{2ij}^{k}}{1 + xa_{2ij}^{k} - xa_{1ij}^{k}}
\end{aligned} \tag{2}$$

Step 3: Compute Total Normalized Crisp Values

$$x_{ij}^k = \frac{xls_{ij}^k\left(1 - xls_{ij}^k\right) + \left(xrs_{ij}^k \times xrs_{ij}^n\right)}{1 - xls_{ij}^k + xrs_{ij}^k} \tag{3}$$

Step 4: Compute the Crisp Values

$$\widetilde{\omega}_{ij}^k = \min a_{ij}^n + \left[x_{ij}^n \times \left(\max r_{ij}^n - \min l_{ij}^n\right)\right] \tag{4}$$

Step 5: Integrate Crisp Values from k Different Respondents

$$\widetilde{\omega}_{ij}^k = \frac{\widetilde{\omega}_{ij}^1 + \widetilde{\omega}_{ij}^2 + \ldots + \widetilde{\omega}_{ij}^k}{k} \tag{5}$$

### 3.2. DEMATEL

The DEMATEL method is an MCDM (multi-criteria decision making) technique developed by the Battelle Memorial Institute's Geneva Research Center in 1972 [74]. The DEMATEL method outperforms other MCDM methodologies (BWM, ISM, TISM, SEM, TOPSIS) in that it reveals the causal relationships between criteria, ranks the criteria according to the type of relationship, and quantifies the intensity of each criterion's effect [75,76]. Furthermore, the DEMATEL threshold is usually determined by experts, so it does not necessarily require a large amount of data [77]. Hence, DEMATEL is used in this study to unveil the causal relationships of Industry 4.0 adoption barriers, rank the barriers based on the net value of influence and affected degree, and visualize the structure of complicated causal relationships. The five-step procedure of DEMATEL is as follows:

Step 1: Generating the Initial Direct Relationship Matrix

The construction of an initial direct relation matrix A, a n × n matrix obtained by pair-wise comparisons, is required to evaluate the relationship between criteria. In the first step, experts posing the required knowledge and experience were asked to indicate the degree of influence each factor *i* exerts on each factor *j*, as indicated by *a_ij*. However, when *i = j*, diagonal elements take zero as their value (*a_ij* = 0), which represents there is no effect.

$$A = \begin{bmatrix} 0 & a_{12} & \cdots & a_{1j} & \cdots & a_{1n} \\ a_{21} & 0 & \cdots & a_{2j} & \cdots & a_{2n} \\ \vdots & \vdots & \vdots & \vdots & \vdots & \vdots \\ a_{i1} & a_{i2} & \cdots & a_{ij} & \cdots & a_{in} \\ a_{n1} & a_{n2} & \cdots & a_{nj} & \cdots & 0 \end{bmatrix}$$

Step 2: Normalizing the Direct Relationship Matrix

In this second step, the initial direct relationship matrix *A* developed in step 1 is normalized using formula (6). The obtained normalized direct relationship matrix is denoted by *S*.

$$S = A \times \frac{1}{\max\limits_{1 \leq i \leq n} \sum_{j=1}^{n} a_{ij}} \tag{6}$$

Step 3: Constructing the Total Relationship Matrix

The normalized direct relationship matrix *S* is converted into the total relationship matrix *T* in this step, using formula (7), in which *I* denotes the identity matrix. Each element of the total relationship matrix ($t_{ij}$), indicates the effects of *i* criterion has on the *j* criterion. Hence, the total relation matrix (*T*) provides an overall relationship between each pair of criteria.

$$T = S(I - S)^{-1} \tag{7}$$

where, $T = [t_{ij}]_{n \times n}$, *i, j* = 1, 2, . . . , *n*.

Step 4: Determining Causal Parameters *D* and *R*

The sum of the rows and the columns within the total relationship matrix $T$ are denoted as $D$ (influence degree) and $R$ (affected degree) and are computed using formulas (8) and (9), respectively. $D$ denotes the degree of direct and indirect influence exerted by criterion $i$ to all other criteria, while $R$ denotes the affected degree of criterion $i$ by all other criteria. Hence, the D-R value shows the net effect of criterion $i$ on the whole system. If the value of D-R is positive, criterion $i$ is clustered under the cause group, while if the value of D-R is negative, criterion $i$ is clustered under the effect group. In practice, criteria under the cause group are usually called critical criteria because they influence the effect group criteria. Moreover, according to Seker and Zavadskas [76], improving cause criteria enhances the effect criteria simultaneously. Therefore, considering the interdependence among factors, much attention should be paid to the cause group criteria compared to the effect group criteria.

$$D = \left[ \sum_{j=1}^{n} t_{ij} \right]_{n \times 1} \tag{8}$$

$$R = \left[ \sum_{i=1}^{n} t_{ij} \right]_{n \times 1} \tag{9}$$

Step 5: Creating a Causal Diagram

In this final step, a cause-and-effect relationship diagram can be generated by mapping the datasets $(D + R, D - R)$ in the Cartesian coordinate system using $(D + R)$ as the horizontal axes and $(D - R)$ as the vertical axes, respectively.

### 3.3. Integration of Fuzzy Set and DEMATEL

Fuzzy sets theory and the DEMATEL method are coupled in this study to robustly evaluate the cause-and-effect relationship among barriers to adopting Industry 4.0 in manufacturing industries. In this integrated approach, fuzzy set theory is applied to deal with the vagueness and imprecision involved in the experts' judgments, and DEMATEL is utilized to evaluate interdependent relationships among obstacles and highlight the critical ones through a visual structural diagram. First, experts' influence degree judgment is collected in linguistic scales and turned into fuzzy numbers; then, the resulting fuzzy numbers are converted into specific numerical values (crisp values) for DEMATEL analysis. The analytical procedure of the fuzzy-based DEMATEL model is graphically described in Figure 1.

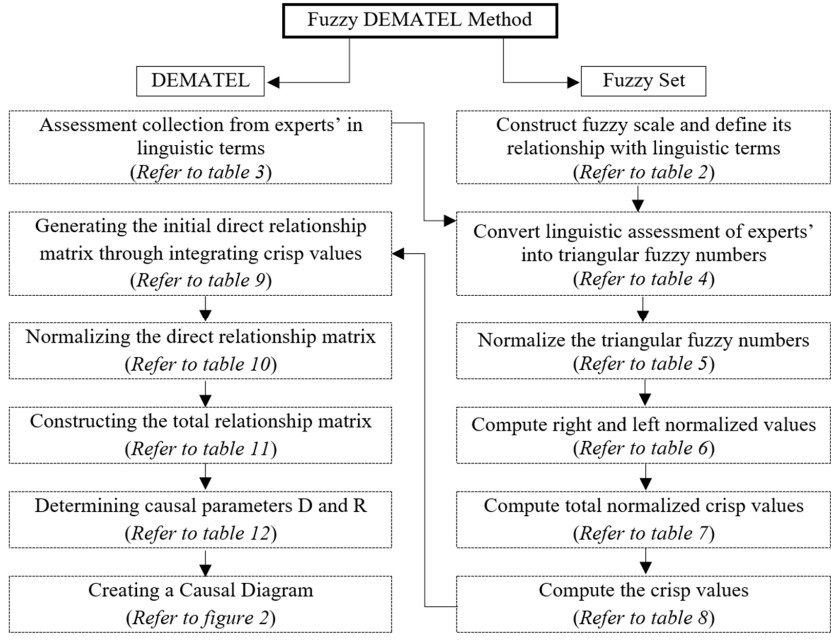

**Figure 1.** Linking of fuzzy set and DEMATEL for this study.

*3.4. Data Collection*

This study utilized a comprehensive review of the literature and expert opinions to determine the adoption barriers of Industry 4.0 in the manufacturing industry. As illustrated in Table 1, there are sixteen challenges to Industry 4.0 adoption that fall into four categories. To collect responses from professionals, a 16 × 16 structural self-interaction matrix (SSIM), that permits pair-wise comparison, was developed. In addition, some basic demographic information (age, years of experience, job position, organization name and highest academic degree) was gathered. Linguistic terms were used to obtain expert evaluations since an ordinal scale (ranked and ordered linguistic variables) is more suitable for expressing preferences [78].

For this study, the comparison scale is divided into five levels: no influence (No), very low influence (VL), low influence (L), high influence (H), and very high influence (VH). Nine experts, with varied ages (35–55 years) and career lengths (10–20 years), who are currently employed as Regional Operations Manager (South Asia), Head of Operations, Chief Research and Innovation Officer, Managing Director, Commercial Planning Director, Strategy and Business Planning Director, and Head of Production in leading manufacturing firms provided their opinions. Before data collection, experts were briefed on the standard definitions and purpose of the study.

**4. Data Analysis**

This study uses an integrated approach of fuzzy sets and the DEMATEL method to assess the cause-and-effect relationship among barriers to Industry 4.0 adoption in manufacturing industries. This combination is utilized to overcome the impreciseness and uncertainty of expert assessment, as well as the benefits of its ability to make valid conclusions from a few expert responses rather than a large number of public responses [79].

Experts were asked to evaluate the relationships among sixteen barriers in a structural self-interaction matrix (SSIM). A sample expert evaluation is shown in Table 3. It is noteworthy that linguistic terms were used to collect expert evaluations. This linguistic assessment is converted into triangular fuzzy numbers as per the defined relationship presented in Table 2, and the resultant outcome is shown in Table 4. Afterward, a five-step CFCS (Converting Fuzzy data into Crisp Scores) defuzzification method is employed to acquire specific values (crisp values) for each expert evaluation. The underlying formulas are comprehensively discussed in Section 3.1. In the first step, each fuzzy triplet (the lower, medium, and upper numbers of the fuzzy sets) is normalized using Formula (1) and compiled in Table 5. In the second step, left and right normalized scores are determined using Formula (2) and recorded in Table 6. Formula (3) is used in the third step to compute total normalized crisp values, as shown in Table 7. Table 8 shows the final crips values obtained in the fourth step using Formula (4). Finally, in a similar procedure, crisp values from all nine expert evaluations are computed. For further analysis, de-fuzzified crisp values from nine experts are aggregated using Formula (5). The resulting matrix is called the initial direct relationship matrix (A), shown in Table 9, indicating the influence each barrier exerts on other barriers. Construction of the initial direct relationship matrix (Table 9) is the first step of DEMATEL, as comprehensively outlined in Section 3.2. The following step normalized the values of the initial direct relationship matrix (A) using Formula (6), and the resulting normalized direct relationship matrix (S) is presented in Table 10. In the third step, with the help of an identity matrix (I), the normalized direct relationship matrix (S) is converted into the total relationship matrix (T) using Formula (7), shown in Table 11.

**Table 3.** Assessment from an expert in linguistic terms.

|  | C1 | C2 | C3 | . . . . . . . . . | C14 | C15 | C16 |
|---|---|---|---|---|---|---|---|
| **C1** | 0 | H | H | . . . . . . . . . | L | H | L |
| **C2** | H | 0 | H | . . . . . . . . . | VL | H | L |
| **C3** | H | H | 0 | . . . . . . . . . | H | H | L |
| ⋮ | ⋮ | ⋮ | ⋮ | ⋮ | . . . . . . . . . | ⋮ | ⋮ | ⋮ |
| **C14** | L | L | L | . . . . . . . . . | 0 | H | H |
| **C15** | VH | VH | L | . . . . . . . . . | L | 0 | H |
| **C16** | No | VL | No | . . . . . . . . . | H | H | 0 |

**Table 4.** Assessment from expert converted into triangular fuzzy numbers.

|  | C1 | C2 | C3 | . . . . . . . . . | C14 | C15 | C16 |
|---|---|---|---|---|---|---|---|
| **C1** | 0 | 0.5, 0.75, 1 | 0.5, 0.75, 1 | . . . | 0.25, 0.5, 0.75 | 0.5, 0.75, 1 | 0.25, 0.5, 0.75 |
| **C2** | 0.5, 0.75, 1 | 0 | 0.5, 0.75, 1 | . . . | 0, 0.25, 0.5 | 0.5, 0.75, 1 | 0.25, 0.5, 0.75 |
| **C3** | 0.5, 0.75, 1 | 0.5, 0.75, 1 | 0 | . . . | 0.5, 0.75, 1 | 0.5, 0.75, 1 | 0.25, 0.5, 0.75 |
| ⋮ | ⋮ | ⋮ | ⋮ | ⋮ | . . . | ⋮ | ⋮ | ⋮ |
| **C14** | 0.25, 0.5, 0.75 | 0.25, 0.5, 0.75 | 0.25, 0.5, 0.75 | . . . | 0 | 0.5, 0.75, 1 | 0.5, 0.75, 1 |
| **C15** | 0.75, 1, 1 | 0.75, 1, 1 | 0.25, 0.5, 0.75 | . . . | 0.25, 0.5, 0.75 | 0 | 0.5, 0.75, 1 |
| **C16** | 0, 0, 0.25 | 0, 0.25, 0.5 | 0, 0, 0.25 | . . . | 0.5, 0.75, 1 | 0.5, 0.75, 1 | 0 |

**Table 5.** Normalized triangular fuzzy numbers.

|  | C1 | C2 | C3 | . . . . . . . . . | C14 | C15 | C16 |
|---|---|---|---|---|---|---|---|
| **C1** | 0 | 0.5, 0.75, 0.75 | 0.5, 0.75, 0.75 | . . . | 0.25, 0.5, 0.5 | 0.5, 0.75, 0.75 | 0.25, 0.5, 0.5 |
| **C2** | 0.5, 0.75, 0.75 | 0 | 0.5, 0.75, 0.75 | . . . | 0, 0.25, 0.25 | 0.5, 0.75, 0.75 | 0.25, 0.5, 0.5 |
| **C3** | 0.5, 0.75, 0.75 | 0.5, 0.75, 0.75 | 0 | . . . | 0.5, 0.75, 0.75 | 0.5, 0.75, 0.75 | 0.25, 0.5, 0.5 |
| ⋮ | ⋮ | ⋮ | ⋮ | ⋮ | . . . | ⋮ | ⋮ | ⋮ |
| **C14** | 0.25, 0.5, 0.5 | 0.25, 0.5, 0.5 | 0.25, 0.5, 0.5 | . . . | 0 | 0.5, 0.75, 0.75 | 0.5, 0.75, 0.75 |
| **C15** | 0.75, 1, 0.75 | 0.75, 1, 0.75 | 0.25, 0.5, 0.5 | . . . | 0.25, 0.5, 0.5 | 0 | 0.5, 0.75, 0.75 |
| **C16** | 0, 0, 0 | 0, 0.25, 0.25 | 0 | . . . | 0.5, 0.75, 0.75 | 0.5, 0.75, 0.75 | 0 |

**Table 6.** Left and right normalized values.

|  | C1 | C2 | C3 | . . . . . . . . . | C14 | C15 | C16 |
|---|---|---|---|---|---|---|---|
| **C1** | 0, 0 | 0.6, 0.75 | 0.6, 0.75 | . . . | 0.4, 0.5 | 0.6, 0.75 | 0.4, 0.5 |
| **C2** | 0.6, 0.75 | 0, 0 | 0.6, 0.75 | . . . | 0.2, 0.25 | 0.6, 0.75 | 0.4, 0.5 |
| **C3** | 0.6, 0.75 | 0.6, 0.75 | 0, 0 | . . . | 0.6, 0.75 | 0.6, 0.75 | 0.4, 0.5 |
| ⋮ | ⋮ | ⋮ | ⋮ | ⋮ | . . . | ⋮ | ⋮ | ⋮ |
| **C14** | 0.4, 0.5 | 0.4, 0.5 | 0.4, 0.5 | . . . | 0, 0 | 0.6, 0.75 | 0.6, 0.75 |
| **C15** | 0.8, 1 | 0.8, 1 | 0.4, 0.5 | . . . | 0.4, 0.75 | 0, 0 | 0.6, 0.75 |
| **C16** | 0, 0 | 0.2, 0.25 | 0, 0 | . . . | 0.6, 0.75 | 0.6, 0.75 | 0, 0 |

**Table 7.** Total normalized crisp values.

|  | C1 | C2 | C3 | . . . . . . . . . | C14 | C15 | C16 |
|---|---|---|---|---|---|---|---|
| **C1** | 0 | 0.698 | 0.698 | . . . | 0.445 | 0.698 | 0.445 |
| **C2** | 0.698 | 0 | 0.698 | . . . | 0.212 | 0.698 | 0.445 |
| **C3** | 0.698 | 0.698 | 0 | . . . | 0.698 | 0.698 | 0.445 |
| ⋮ | ⋮ | ⋮ | ⋮ | ⋮ | . . . | ⋮ | ⋮ | ⋮ |
| **C14** | 0.445 | 0.445 | 0.445 | . . . | 0 | 0.698 | 0.698 |
| **C15** | 0.967 | 0.967 | 0.445 | . . . | 0.445 | 0 | 0.698 |
| **C16** | 0 | 0.212 | 0 | . . . | 0.698 | 0.698 | 0 |

**Table 8.** Crisp values.

|  | **C1** | **C2** | **C3** | . . . . . . . . . | **C14** | **C15** | **C16** |
|---|---|---|---|---|---|---|---|
| **C1** | 0 | 0.698 | 0.698 | . . . | 0.445 | 0.698 | 0.445 |
| **C2** | 0.698 | 0 | 0.698 | . . . | 0.212 | 0.698 | 0.445 |
| **C3** | 0.698 | 0.698 | 0 | . . . | 0.698 | 0.698 | 0.445 |
| ⋮ | ⋮ | ⋮ | ⋮ | ⋮ | . . . | ⋮ | ⋮ | ⋮ |
| **C14** | 0.445 | 0.445 | 0.445 | . . . | 0 | 0.698 | 0.698 |
| **C15** | 0.967 | 0.967 | 0.445 | . . . | 0.445 | 0 | 0.698 |
| **C16** | 0 | 0.212 | 0 | . . . | 0.698 | 0.698 | 0 |

**Table 9.** Initial direct relationship matrix (A).

|  | **C1** | **C2** | **C3** | . . . . . . . . . | **C14** | **C15** | **C16** |
|---|---|---|---|---|---|---|---|
| **C1** | 0 | 0.819 | 0.654 | . . . | 0.619 | 0.672 | 0.673 |
| **C2** | 0.759 | 0 | 0.676 | . . . | 0.679 | 0.819 | 0.644 |
| **C3** | 0.731 | 0.759 | 0 | . . . | 0.705 | 0.673 | 0.531 |
| ⋮ | ⋮ | ⋮ | ⋮ | ⋮ | . . . | ⋮ | ⋮ | ⋮ |
| **C14** | 0.761 | 0.759 | 0.673 | . . . | 0 | 0.622 | 0.701 |
| **C15** | 0.701 | 0.789 | 0.644 | . . . | 0.701 | 0 | 0.700 |
| **C16** | 0.566 | 0.618 | 0.431 | . . . | 0.877 | 0.789 | 0 |

**Table 10.** Normalized direct relationship matrix (S).

|  | **C1** | **C2** | **C3** | . . . . . . . . . | **C14** | **C15** | **C16** |
|---|---|---|---|---|---|---|---|
| **C1** | 0 | 0.082 | 0.066 | . . . | 0.062 | 0.067 | 0.067 |
| **C2** | 0.076 | 0 | 0.068 | . . . | 0.068 | 0.082 | 0.064 |
| **C3** | 0.073 | 0.076 | 0 | . . . | 0.071 | 0.067 | 0.053 |
| ⋮ | ⋮ | ⋮ | ⋮ | ⋮ | . . . | ⋮ | ⋮ | ⋮ |
| **C14** | 0.076 | 0.076 | 0.067 | . . . | 0 | 0.062 | 0.070 |
| **C15** | 0.070 | 0.079 | 0.066 | . . . | 0.070 | 0 | 0.070 |
| **C16** | 0.057 | 0.062 | 0.043 | . . . | 0.088 | 0.079 | 0 |

**Table 11.** Total relationship matrix (T).

|  | **C1** | **C2** | **C3** | . . . . . . . . . | **C14** | **C15** | **C16** |
|---|---|---|---|---|---|---|---|
| **C1** | 0.754 | 0.827 | 0.747 | . . . | 0.748 | 0.730 | 0.712 |
| **C2** | 0.844 | 0.771 | 0.766 | . . . | 0.772 | 0.760 | 0.727 |
| **C3** | 0.827 | 0.827 | 0.691 | . . . | 0.760 | 0.733 | 0.703 |
| ⋮ | ⋮ | ⋮ | ⋮ | ⋮ | . . . | ⋮ | ⋮ | ⋮ |
| **C14** | 0.838 | 0.835 | 0.759 | . . . | 0.703 | 0.739 | 0.726 |
| **C15** | 0.829 | 0.834 | 0.754 | . . . | 0.765 | 0.676 | 0.723 |
| **C16** | 0.738 | 0.741 | 0.662 | . . . | 0.709 | 0.681 | 0.591 |

The entire relationship matrix (T) indicates the total influence, both direct and indirect, holistically posited by each barrier on other barriers. Next, using Formulas (8) and (9), the sum of rows and columns is calculated within the total relationship matrix (T). The sum of the rows and columns, as shown in Table 12, is presented by D (influence degree) and R (affected degree), respectively. Finally, based on the D-R values (net effect), barriers are categorized into cause-and-effect groups and ranked in the priority hierarchy.

**Table 12.** Comprehensive influence scores.

|  | Influence Degree (D) | Affected Degree (R) | Centrality (D + R) | Cause Degree (D − R) |
|---|---|---|---|---|
| C1 | 11.221 | 12.517 | 23.738 | −1.296 |
| C2 | 11.511 | 12.469 | 23.980 | −0.958 |
| C3 | 11.288 | 11.382 | 22.669 | −0.094 |
| C4 | 10.217 | 10.825 | 21.042 | −0.608 |
| C5 | 11.135 [2] | 10.660 | 21.795 | 0.475 [4] |
| C6 | 10.783 [3] | 9.018 | 19.801 | 1.765 [1] |
| C7 | 10.582 [4] | 9.321 | 19.903 | 1.262 [2] |
| C8 | 10.215 [5] | 9.856 | 20.071 | 0.359 [5] |
| C9 | 9.719 [6] | 8.971 | 18.691 | 0.748 [3] |
| C10 | 9.853 | 10.371 | 20.224 | −0.518 |
| C11 | 9.132 | 9.417 | 18.549 | −0.285 |
| C12 | 10.999 | 11.412 | 22.410 | −0.413 |
| C13 | 11.676 | 11.794 | 23.470 | −0.117 |
| C14 | 11.414 | 11.452 | 22.867 | −0.038 |
| C15 | 11.364 [1] | 11.067 | 22.431 | 0.297 [6] |
| C16 | 10.191 | 10.768 | 20.959 | −0.577 |

Finally, a cause-and-effect relationship diagram, shown in Figure 2, is generated in the fifth step by plotting (D + R) in the abscissa and (D − R) in the ordinate, respectively.

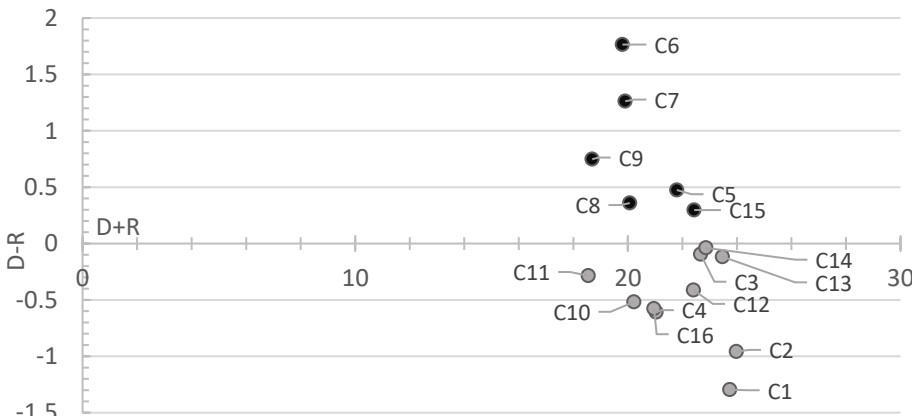

**Figure 2.** Cause-and-effect relationship diagram.

### 4.1. Grouping of Barriers and Their Significance

As shown in Figure 2, sixteen barriers to implementing Industry 4.0 in manufacturing organizations have been grouped into cause-and-effect categories based on net effect (D-R) value. Among the sixteen barriers depicted in Table 12, six barriers that belong to the cause group in the descending order of importance are C6: requirement for high initial investment (1.765), C7: uncertainty of return-on-investment (1.262), C9: government support and legal issues (0.748), C5: capability to manage big data (0.475), C8: availability of reference architecture and standards (0.359), and C15: availability of skilled workforce (0.297). According to the DEMATEL principle, these six hurdles, which have a positive net value, are the most critical barriers to adopting Industry 4.0 in manufacturing enterprises. On the other hand, the obstacles that belong to the effect group are 'technology availability and compatibility' (C1), 'low maturity of technology and seamless integration' (C2), 'information technology infrastructure' (C3), 'cyber-security and privacy' (C4), 'complexity in supply chain integration and coordination' (C10), 'employee fear and resistance to change' (C11), 'education and training programs' (C12), 'knowledge, awareness, and competence of I4.0' (C13), 'management commitment and leadership' (C14), and 'organization structure and culture' (C16). Moreover, as reported by Wan et al. [75] (2021), the cause group barriers

have a substantial direct and indirect influence on the elements of the effect group. Hence, the effect group's hurdles are not deemed critical and must be handled after the cause group's barriers are addressed.

In terms of how barriers are linked, the most critical hurdles (cause group barriers) are found to be distributed in the economic dimension, which includes the 'requirement for high initial investment' (C6) and 'uncertainty of return-on-investment' (C7). In comparison, the outcome barriers (effect group barriers) are primarily technological in nature, consisting of 'technology availability and compatibility' (C1), 'low maturity of technology and seamless integration' (C2), 'information technology infrastructure' (C3), and 'cyber-security and privacy' (C4), with the exception of (C5) 'capability to manage big data' which falls in the cause group. Additionally, the effect group barriers dominate both regulatory and organization dimensions, as seen in Table 1. This distribution of barriers reveals that economic obstacles are the major stumbling block to Industry 4.0 adoption in manufacturing industries due to their direct impact on technological, regulatory, and organizational aspects.

Some of the findings of this study are consistent with previous research, while others provide new insight. As shown in this study, 'requirement for high initial investment' [29,30,35,39,40,46,51,80] and 'uncertainty of return-on-investment' [29,35,43,51,60] are also found to be the most prominent obstacles to adopting Industry 4.0 in different geographic locations.

In line with the result of this study, Bakhtari et al. [35], Chauhan et al. [39], Stentoft et al. [40], Obiso et al. [43], and Narwane et al. [60] also highlighted 'availability of skilled workforce' as a crucial barrier to Industry 4.0 implementation. In addition, Horváth and Szabó [29], Kamble et al. [30], Bakhtari et al. [35], Chauhan et al. [39], Stentoft et al. [40], Obiso et al. [43], Moktadir et al. [46], and Aggarwal et al. [51] reported 'capability to manage big data' as a major difficulty in adopting Industry 4.0. Finally, Horváth and Szabó [29], Kamble et al. [30], Bakhtari et al. [35], Stentoft et al. [40], and Obiso et al. [43] also listed 'availability of reference architecture and standards' as one of the major impediments to Industry 4.0 adoption. Even though it is rarely mentioned in the literature, this study identifies 'government support and legal issues' as a critical hurdle to implementing Industry 4.0.

### 4.2. Presentation of Causal Relationship among Critical Barriers

Based on the total relationship matrix (T), a causal relationship diagram (Figure 3) has been developed to visualize the complex association among the critical barriers to Industry 4.0 adoption. The diagram is constructed based on the overall relationship values (as given in Table 11) of the six crucial adoption barriers identified in the preceding subsection to demonstrate the effect group barriers and the directions of impacts. This causal relationship diagram incorporates twelve out of the sixteen barriers considered in this study. However, for ease of conceptualization and interpretation, the abovementioned diagram only includes the effect group barriers and their respective directions that have a very high degree of influence (>0.75). So, it does not imply that no additional relationship exists among the barriers; instead, the other associations are not very strong on the assessment scale used in this study, as illustrated in Table 2.

From the causal relationship diagram shown in Figure 3, it is evident that the identified critical barriers have a substantial impact on the six adoption barriers, which include 'technology availability and compatibility' (C1), 'low maturity of technology and seamless integration' (C2), 'information technology infrastructure' (C3), 'education and training programs' (C12), 'knowledge, awareness, and competence of Industry 4.0' (C13), and 'management commitment and leadership' (C14). Among the effect group barriers, 'technology availability and compatibility' (C1) has a strong correlation with each of the six critical obstacles. Whereas 'education and training programs' (C12) and 'government support and legal issues' (C9) are strongly connected solely with 'lack of availability of skilled workforce' (C15) and 'technology availability and compatibility' (C1), respectively. Moreover, 'requirement for high initial investment' (C6), the most critical barrier to Industry 4.0 adoption in the manufacturing industries, found to have a significant influence on the 'technology

availability and compatibility' (C1), 'low maturity of technology and seamless integration' (C2), and 'information technology infrastructure' (C3). Moreover, as depicted in the figure, 'low maturity of technology and seamless integration' (C2), and 'knowledge, awareness, and competence of Industry 4.0' (C13) have a bidirectional link with the 'lack of availability of skilled workforce' (C15).

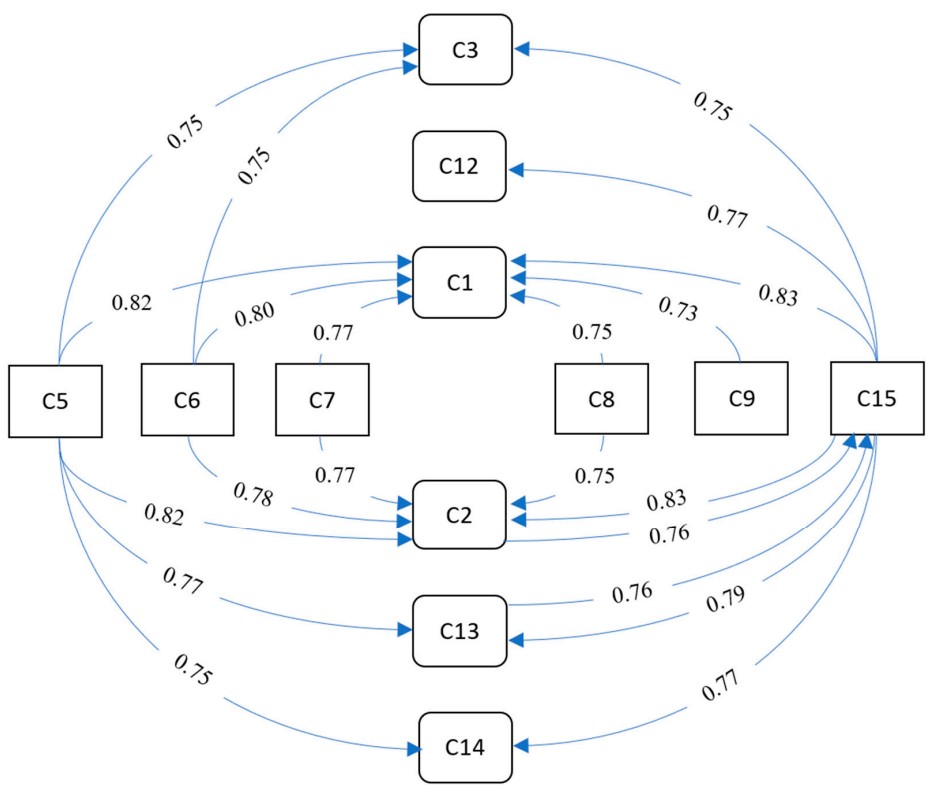

**Figure 3.** Causal relationship among critical barriers of Industry 4.0 adoption.

In a fragmented fashion, the literature also reported similar correlations among the barriers to Industry 4.0 adoption. For instance, in line with the findings of this study, Chauhan et al. [39] also suggested that active government initiatives to support the required technologies could ease Industry 4.0 adoption. They also reported that the lack of legal support through policy adversely affected the digitalization of businesses in developing countries. Moreover, Ozkan-Ozen and Kazancoglu [81] revealed a connection between specialized training programs and filling the labor market's workforce shortage with the right skillset. Similarly, Mian et al. [82] emphasize the significance of reshaping higher education following the vision of Industry 4.0 to equip novices with the requisite skills.

## 5. Mapping of Critical Barriers and Proposing Mitigation Strategies

*5.1. Mapping of Critical Barriers to Industry 4.0 Adoption in Different Economies*

To answer the second research question of this study, a three-step approach has been considered as follows.

(i)     Selection of suitable articles:

The relevant articles from journals listed in the Web-of-Science database were extracted using the purposive sampling technique [83]. At first, we searched the journal articles with the keywords "Industry 4.0" and "barriers" in the title of the article. Second, we limited the search using the criteria: "geography/region" and "year of study 2018–2022", and obtained 35 articles. Third, this comparison study targeted only those peer-reviewed articles that discussed the critical barriers identified and the adoption challenges considering the country/geographic location perspective. Eventually, the result of this search and read

provides three potential articles: Bravi and Murmura [84], Raj et al. [41], and Fernando et al. [85]. Among these articles, the first two are grounded on evidence from Italian and French industries, and the rest encompassed the data from the largest economy in Southeast Asia, one of the emerging market economies in Asia.

(ii)   Mapping criterion:

The inclusiveness of all the barriers recognized as critical for both developed and developing countries was set as a mapping criterion for this study. The six critical obstacles identified in this study were considered a starting point to enter into this mapping. Subsequently, we include barriers that were deemed critical in other selected articles. This mapping allows us to compare the existence and importance of barriers explored in diverse economic and geographical contexts.

(iii)   Illustration of comparative analysis:

Figure 4 displays the mapping of critical adoption barriers according to the articles studied. Interesting to note here that ten out of twelve prevailing barriers matched within our initial set (sixteen) of barriers. As shown in Figure 4, the most significant hurdle to Industry 4.0 implementation identified in this study is the requirement for a substantial initial investment. For Indonesian manufacturing companies, the biggest challenge is the government's unclear policy on this issue. Since the Industry 4.0 adoption effort is made solely at the corporate level in countries such as Indonesia and other similar economies, high capital investment requirements, lack of funding partners, and obscure government policy discourage business owners and top management from investing in and innovating to overcome other Industry 4.0 adoption barriers. Hence, the government should frame detailed guidelines and appropriate legislation on funding programs, tax benefits, and other incentives for manufacturing firms to help mitigate the afforested obstacles. Again, low technological maturity and the challenge of finding innovation cooperation partners to obtain the necessary support for Industry 4.0 implementation are noted as the most significant barriers for the French and Italian manufacturing organizations, respectively. However, these hurdles are often solved through organizational efforts in industrialized countries such as France and Italy, as these countries have well-defined national strategies and policies to encourage companies to adopt Industry 4.0 technologies and promote innovative partners, e.g., universities or start-ups to work collaboratively with business organizations and provide the guidance and support they need.

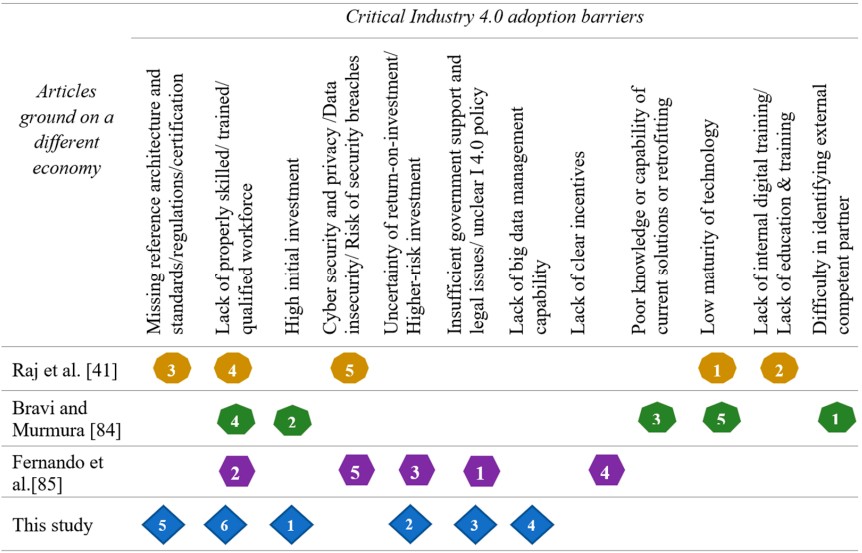

| Articles ground on a different economy | Critical Industry 4.0 adoption barriers | | | | | | | | | | | |
|---|---|---|---|---|---|---|---|---|---|---|---|---|
| | Missing reference architecture and standards/regulations/certification | Lack of properly skilled/trained/qualified workforce | High initial investment | Cyber security and privacy/Data insecurity/Risk of security breaches | Uncertainty of return-on-investment/Higher-risk investment | Insufficient government support and legal issues/unclear I 4.0 policy | Lack of big data management capability | Lack of clear incentives | Poor knowledge or capability of current solutions or retrofitting | Low maturity of technology | Lack of internal digital training/Lack of education & training | Difficulty in identifying external competent partner |
| Raj et al. [41] | 3 | 4 | | 5 | | | | | | | 1 | 2 |
| Bravi and Murmura [84] | | 4 | 2 | | | | | | 3 | 5 | | 1 |
| Fernando et al. [85] | | 2 | | | 5 | 3 | 1 | | 4 | | | |
| This study | 5 | 6 | 1 | | | 2 | 3 | 4 | | | | |

**Figure 4.** Mapping of critical adoption barriers.

Again, this study finds the uncertainty of return on investment as the second most critical challenge to implementing Industry 4.0, whereas it is the scarcity of workforce with appropriate skills for Indonesian manufacturing organizations. Given that the Industry 4.0 implementation effort is so far made exclusively by the organizations themselves, manufacturing companies in Indonesia and other similar economies show little interest in investing in such a high-cost initiative combined with a high risk of failure unless the firms have a clear picture of the economic benefit on investment and readily available workforce with the right qualifications. However, with the increasing shift from certificate-based to skill-based hiring in the job market, employees from different professions and new job seekers are now being impelled to equip themselves with the required hard and soft skills. This job market push, in turn, is driving vocational schools and technical universities to modify their curricula in line with the upskilling of Industry 4.0 and accommodate the upskill programs on digitalization. Hence, it can be predicted that the qualified workforce will no longer be considered a barrier to Industry 4.0 adoption shortly. Again, French manufacturing firms see the lack of internal digital training as their second most significant impediment to adopting Industry 4.0. This deficiency arises from missing training concepts and inadequate interdisciplinary courses in technical universities' curricula. As a result, business organizations in industrialized countries focus on establishing training facilities to prepare their workforce with the hard skills linked to Industry 4.0. In contrast, being a part of developed economies, Italian manufacturing organizations, particularly small and medium enterprises (SMEs), surprisingly considered the high initial investment as their second most crucial constraint to adopting Industry 4.0.

Moreover, Figure 4 shows that the missing reference architecture and standards are the hurdle to Industry 4.0 implementation, which is typical for this study and French manufacturing companies. These issues usually result from a lack of global standards, incompatible ports and interfaces, and unstandardized data formats. However, the availability of a dynamic implementation plan with well-defined standards is critical as it encourages managers, entrepreneurs, and business houses to implement, invest and innovate to overcome the challenges in adopting Industry 4.0. A successful and profit-making model/standard/certification, in fact, acts as a catalyst for the organizations planning to implement Industry 4.0.

In addition, the Industry 4.0 adoption barrier commonly seen in all of the studies being compared is the deficiency of a qualified workforce. However, the order of its significance is not the same: Indonesian manufacturing companies consider it the second most critical barrier to overcome, whereas French and Italian manufacturing organizations see it as the fourth Industry 4.0 adoption barrier. Surprisingly, our study finds the lack of workers with appropriate skills as a significant but last (sixth) hurdle to overcome for implementing Industry 4.0. Apart from this, the availability of qualified and skilled personnel at a reasonable cost motivates business organizations to adopt new technology or process. Thus, the industry association, academia, and government must work together to address this issue.

Moreover, for implementing Industry 4.0, the Indonesian and French manufacturing organizations consider the risk of security breaches a critical issue that must be addressed. In fact, no firm can grow if it does not protect its trade secrets or maintain its data privacy. Again, data security requires the availability of compatible encryption technology and a solid regulatory framework, the absence of which is considered a crucial constraint to Industry 4.0 implementation for Indonesian and our manufacturing organizations. Hence, the governments should act quickly to create a legislative framework supporting the successful deployment of Industry 4.0. Again, the Industry 4.0 adoption requires expanded connectivity among the value chain partners that generate a large amount of data. The complexity and heterogeneity of data make them challenging to store, synchronize, and real-time sharing of quality data in terms of completeness, consistency, and accuracy. In addition, the lack of sophisticated data analysis techniques that allow large-scale problems to be visualized, verified, and optimized in dynamic settings makes Industry 4.0 adoption

more difficult for businesses [86]. For these reasons, in our study, the lack of big data management competence has emerged as a critical impediment that needs to be overcome for implementing Industry 4.0.

From Figure 4, it is also evident that, unlike French and Italian manufacturing companies, insufficient government support/unclear I 4.0 policy is a significant obstacle for the Indonesian and studied manufacturing firms willing to implement Industry 4.0. Moreover, though the order of importance is not the same, French and Italian manufacturing industries think low technological maturity is a significant hurdle to be overcome for implementing Industry 4.0. However, it is not a challenge that Indonesians or our manufacturing enterprises must overcome.

The findings also identified that some barriers existed only in emerging economies. These factors included uncertainty of return on investment, insufficient government support, lack of big data management capability, and lack of clear incentives. Among them, the first two factors were found common in the aforestated economy. On its counterpart, the only prevalent Industry 4.0 adoption hurdles are low maturity of technology, poor knowledge/capability of current technological solutions, lack of education and training, and difficulty in identifying competent external partners.

### 5.2. Strategies to Mitigate the Adoption Barriers to Industry 4.0

To overcome critical barriers mapped above against those of the different economies and support the activities necessary to fulfill the vision of industrial revitalization via Industry 4.0, mitigation strategies have been developed. In this context, a focus group discussion (FGD) was conducted with the five industrial experts (selected from the nine participants of the study's survey). Based on the opinions extracted from the discussion and literature evidence, the strategies have been formulated and presented in Figure 5.

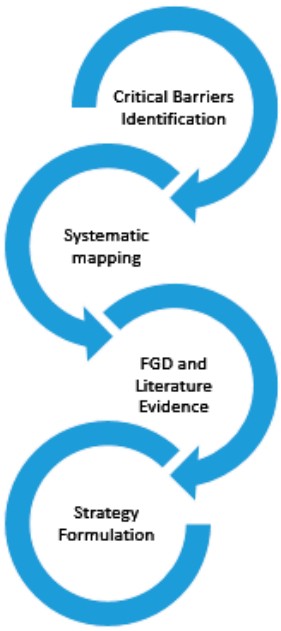

**Figure 5.** Illustration of strategy formulation steps.

### 5.2.1. Mitigation Strategy to Overcome the Critical Economic Barriers Identified (C 6: Requirement for High Initial Investment, and C 7: Uncertainty of Return-on-Investment)

As a technology-driven "Manufacturing Renaissance," implementing Industry 4.0 requires a considerable capital investment. Therefore, business owners and managers should have adequate funds to overcome the financial barrier to adopting Industry 4.0. This high investment needed at the early stages of adopting Industry 4.0 is the most critical hurdle for companies operating in developing countries. In this context, the government should take the initiative to provide necessary financial support as the private financing

partners often do not intend to invest in high-risk unorthodox initiatives such as Industry 4.0. Aside from that, other steps, such as lowering taxes on purchasing Industry 4.0 technology and granting a tax credit for research and development investments, could be adopted to overcome this financial obstacle [87]. In addition, as proof of concept, the government should present business cases demonstrating a substantial Return On Investment (ROI) following the implementation of Industry 4.0 [88]. This strategy would, in turn, help (i) reduce the uncertainty associated with profitability and (ii) attract external private investment partners.

Moreover, as the Industry 4.0 initiative takes time to pay off [55], businesses should plan earlier for their capital resources' acquisition, allocation, and expenditure. Furthermore, to alleviate the initial high capital investment burden, particularly for SMEs, Industry 4.0 adoption should be regarded as a series of discrete projects rather than a massive, company-wide change effort. A reasonable strategy could be to start with a few essential products and services that directly impact productivity, revenue, and customer satisfaction and then scale up as improvements and experiences are obtained.

5.2.2. Mitigation Strategy to Overcome the Critical Regulatory Barriers Identified (C 9: Government Support and Legal Issues, and C 8: Availability of Reference Architecture and Standards)

*Government support:* Significant technological advances may raise important public policy concerns. In its report, McKinsey [89] emphasizes that governments' ability to adopt the appropriate policies is critical to successful adaptation to emerging technology conditions. Governments unable to implement the necessary long-term policies jeopardize their economy; when all other economies are functioning at a rapid pace, their inability to adjust to the new business environment leads to a decline in their competitiveness. Therefore, a government-led initiative such as Advanced Manufacturing Partnership (USA), Manufacture Innovation 3.0 (Korea), and Production 2030 (Sweden), should be developed with a focus on critical national manufacturing sectors and their capabilities and implemented to consolidate global competitiveness in manufacturing via Industry 4.0.

Another critical strategy that the government should pursue is to play a pivotal role in developing a triangular industry–government–academia partnership in areas such as technology development and exchange, need-based training, and collaborative establishment of R&D laboratories or institutions [90]. In addition, the government should encourage the development of an interconnected regional network of industries to expedite the sharing of innovative and best technological practices among cross-industrial sectors [91,92]. Establishing collaboration with overseas companies executing Industry 4.0 would enable domestic organizations to learn from leading examples and create a flow of best practices. Finally, the government should also provide a one-stop service to help the business community mitigate the obstacles at various stages of Industry 4.0 implementation, such as purchasing, installing, and maintaining Industry 4.0 technologies, establishing integration across value chain partners, and failure risks, obtaining technical and regulatory support and educating and training workforces.

*Regulatory framework/legal issues:* Like every other significant technological advancement, the new production processes linked to Industry 4.0 confront the established regulatory framework, posing two interrelated issues. First, uncertainty about the legality of emerging technology or associated data protection and liability concerns stifles its acceptance and impedes the process of innovation. Second, it becomes challenging to enforce current laws due to the de facto dominance of emerging technology and business models. Therefore, two things must be carried out to reconcile regulation with technology: criteria must be developed to ensure that emerging technologies adhere to the law, and a regulatory framework must be constructed to promote innovation. In this context, the government should take the following measures to address the regulatory challenges for the Industry 4.0 program to succeed: (i) developing and enforcing legislation to ensure data security and the requirement to provide documentary evidence when transferring goods from one partner to another (to tackle liability issues); (ii) ensuring legally that Industry 4.0 requirements do

not violate employees' rights to data protection; (iii) making self-regulation through audits or compliance with the standards of IT security a legal requirement; and (iv) developing a legal framework to promote engagement of engineers and legal professionals from the very beginning of R&D processes. In addition, as echoed in some previous work (e.g., Kagermann et al. [7]), SME-specific practical guidelines, checklists, and contract clauses should be developed to safeguard business and trade secrets and ensure the equitable sharing of the value contributed by the new business models.

*Availability of reference architecture and standards:* The reference architecture refers to the detailed technical description and adoption of the standards outlining the collaboration mechanisms and the information to be shared across different corporate entities, as Industry 4.0 allows for inter-company connectivity and consolidation via value networks. It is thus a generic model that serves as a broad framework for configuring, developing, assimilating, and running Industry 4.0-related technological systems and is made available in the form of software applications and services. Nonetheless, creating a reference architecture faces a significant challenge in integrating the perspectives that currently exist in different technological domains pertinent to Industry 4.0 and building a unified approach. Hence, to outline the critical points of standardization and a reference architecture, a 'Working Group' should be formed with the support of the Industry 4.0 platform. Then appropriate flagship projects should be developed to implement the reference architecture successfully; such a suggestion is in line with Kagermann et al. [7]. In addition, maturity assessment models should be developed for regional industries to evaluate their readiness and to monitor and benchmark their progress as they implement Industry 4.0.

### 5.2.3. Mitigation Strategy to Overcome the Critical Technological Barrier Identified (C 5: Capability to Manage Big Data)

Industry 4.0 requires the end-to-end digital integration of entire supply chain partners. With the adoption of Industry 4.0 in business organizations, the use of cyber-physical systems (CPS) grows, necessitating the development of an IT infrastructure capable of sharing data at a significantly greater volume and higher quality than existing communication networks. Therefore, the current communication networks should be upgraded to assure high operational reliability, low latency, widely accessible bandwidth, cost-effective worldwide roaming, quality of service, and widespread availability of SIM cards with embedded chips [7]. With this in mind, the government should invest more in expanding the infrastructure required for high-speed internet access throughout the country. Moreover, a robust and resilient cyber security system should be developed, and proper initiatives should be taken to launch it collaboratively throughout the entire manufacturing value chain to ensure cyber security [93]. Finally, an appropriate legal and regulatory framework for managing data privacy, data ownership, and copyrights should be constructed and efficiently implemented to overcome the barriers to managing big data, cyber-security, and data privacy.

### 5.2.4. Mitigation Strategy to Overcome the Critical Organizational Barrier Identified (C 15: Availability of Skilled Workforce)

Developing technical skill sets in the current and prospective workforce is of utmost importance since having employees with pertinent knowledge and skills is crucial to maximizing the benefits of Industry 4.0. The essential competencies and skills include adaptive thinking, deductive reasoning, and computational skills, particularly in the areas of information technology and data analytics [82]. Likewise, the Industry 4.0 workforce is found to require broad areas of knowledge in information and communication technology, algorithms, automation, software development and security, but also data analysis, a set of skills in leadership, strategic vision, communication, creativity, problem-solving, collaborative work, innovation, adaptability, flexibility, and self-management [94]. It is widely acknowledged that such specialized technical knowledge and skills can be acquired through vocational and academic training as part of continuing professional development and formal education at institutions [82]. Therefore, the government and universities should

join hands to develop programs and curriculums to prepare graduates with the required competencies and skills. Dedicated training facilities, i.e., training hubs, competence centers, and vocational programs, should be launched for the skill development of the current and prospective workforce. In addition, a consultant pool can be developed through certified training and industrial Ph.D. programs [95] (Arlinghaus and Rosca, 2021). Furthermore, business organizations should formulate policies to promote an innovative culture of exchanging ideas and learning new technologies [91]. Finally, embracing the idea reported by Kagermann et al. [7], government and business organizations, together with academic institutions, should launch a joint initiative to meet the demands for new structures of training and content resulting from Industry 4.0, as depicted in Figure 6.

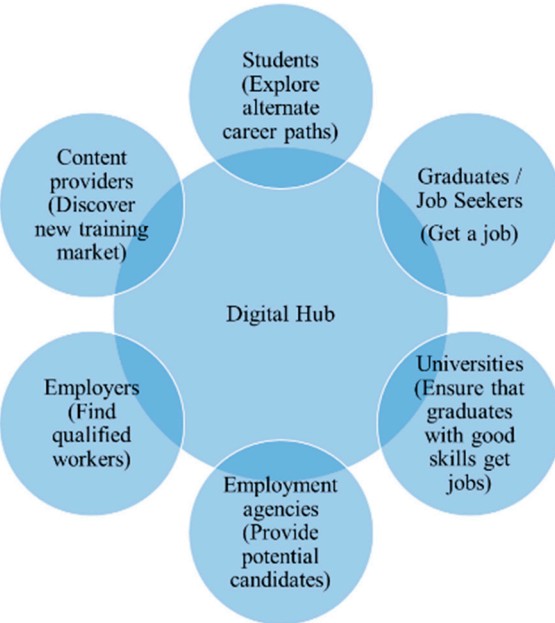

**Figure 6.** Using a digital hub to connect various actors.

To conclude, the strategy to mitigate the adoption barriers to Industry 4.0 is company-specific and should be developed based on the company's vision, core products or services, technical and managerial capabilities, and intended market.

## 6. Conclusions and Implications

This study examined the causal relationship among the critical barriers to Industry 4.0 adoption. This article utilized an integrated fuzzy-DEMATEL approach, preceded by a discussion with industry experts working in globally operated manufacturing firms. This study initially summarizes sixteen barriers, among which six are distinguished as critical. 'Requirement for high initial investment', 'uncertainty of return-on-investment', 'government support and legal issues', 'capability to manage big data', 'availability of reference architecture and standards', and 'availability of skilled workforce' are the essential barriers listed in descending order of importance. Next, a causal diagram holistically demonstrating the influence direction is constructed based on the high degree of influence. This relationship diagram visualizes the all-encompassing relationships among the adoption barriers of Industry 4.0. Furthermore, the most prominent finding from the distribution of obstacles reveals that hurdles connected to the economic dimension play a decisive role in adopting Industry 4.0, particularly in manufacturing industries. This dimension also influences technological, regulatory, and organizational dimensions. It is worth noting that the approach—a combination of fuzzy set theory and the DEMATEL–used in this empirical research was one of the first to look into how the barriers of Industry 4.0 are causally interconnected.

The second goal of this study was to investigate the variations in I4.0 adoption barriers considering various economic orientations and geographical locations. To this extent, this research systematically examined the priority placed on critical barriers in Industry 4.0 implementation among selected setups. One significant finding to emerge from this part of the study was that while high capital investment requirements and unclear government policies on I4.0 were identified as substantial issues primarily for emerging economies, other factors such as low technological maturity and the challenge of identifying competent partners were reported as most critical in the studied article covering the developed economies. Interestingly, some adoption barriers were found critical for manufacturing organizations of emerging economies but not for developed ones, and vice-versa. Furthermore, the most striking observation in all of the studies compared was the lack of a properly trained/qualified workforce. Finally, concerning the third research question, strategies have been developed to mitigate the barriers identified as crucial for Industry 4.0 adoption.

### 6.1. Implications

The authors anticipate that this study's findings will help tackle adoption barriers in both the industry and the literature since they can pave the way for a deeper academic understanding of the mechanisms underlying the barriers and provide critical insights for managers to develop corporate strategies to overcome obstacles. Furthermore, the insights gained from this study would help the decision-makers set where to focus while adopting Industry 4.0 in manufacturing organizations. Moreover, strategies formulated in this study are expected to be helpful for professionals in developing or implementing Industry 4.0.

### 6.2. Limitations and Future Research

Based on the academic literature and practitioner's view, this study focused on critical barriers and their mitigation directions for ensuring the advantages of Industry 4.0. However, this study's limitation lies in how a manufacturing firm can connect the strategies to the competitiveness driven by I4.0 implementation. Thus, further research is required to design a new business model intended to draw the relationship between strategy, competitiveness, and profitability connected to such a recent phenomenon. Moreover, the generalizability of these results is subject to certain limitations. The paper demonstrates the nature and behavior of several barriers to Industry 4.0 in the context of manufacturing industries. For instance, the economic dimension has difficulties adopting Industry 4.0, particularly affecting technological, regulatory, and organizational dimensions. However, the concept of Industry 4.0 is vast, and to obtain countrywide views, such a phenomenon can be investigated by covering other business sectors. Thus, the critical factors might not be equally important for all sectors of an economy. A greater focus on identifying and mapping barriers could produce interesting findings that account more for the sector-independent result of an economy or country.

**Author Contributions:** A.S., M.M.A.K. and P.K.B. contributed to the study conception and design. A.S. and P.K.B. collected related papers for literature review. A.S., M.M.A.K. and P.K.B. conducted the survey and case study to collect data. All authors were involved at different stages of data analysis and strategy formulation. A.S. and M.M.A.K. wrote the first draft of the manuscript. L.R. and M.D.M. contributed to review and developing the final version of this paper. All authors have read and agreed to the published version of the manuscript.

**Funding:** This study received funding from University Research Center, Shahjalal University of Science and Technology (AS/2021/1/34).

**Data Availability Statement:** The data are not publicly available due to the confidential agreement with the funding Institution. However, the data will be provided on request.

**Acknowledgments:** We would like to thank the industrial experts who participated in this research for their insightful feedback and comments on the relationship matrix.

**Conflicts of Interest:** The authors report there are no competing interests to declare.

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
