# Peer review of "Critical Barriers to Industry 4.0 Adoption in Manufacturing Organizations and Their Mitigation Strategies"

_jmmp, doi:10.3390/jmmp6060136_

Round 1

Reviewer 1 Report

The paper presents a critical analysis of existing barriers for the adoption of Industry4.0 in manufacturing, by clustering them according to their dimensions, and further integrating them into a cause-effect diagram. Moreover, grounding on a panel discussion with experts from various sectors, they identify mitigation strategies.
The strong aspects of this work are definitely the sounding methodology used to identify, review and analyze the critical barriers to I4.0. Much space in the paper is devoted to explain such methodology. Though interesting, Section 3 could be synthetized better, for instance moving Tables to the Appendix, since they do not provide critical information for the analysis, which is the added value of the work.
On the other hand, Authors are kindly suggested to motivate a bit more on the possible mitigation actions, clearly linking them to the cause-effect relations identified previously. For instance, Requirement for high initial investment (5.2.1) is identified as first mitigation strategy. This is a barrier rather than a mitigation strategy. Within the paragraph, government intervention is invoked to mitigate this issue (fundings etc). However, the following paragraph focuses on government intervention. Thus, Authors are kindly invited to verify how mitigation strategies are identified, as well as understanding which ones should be endogeneous, i.e. up to the company (e.g. improve training in digital tools for operators?) and which ones should be exogeneous, i.e. up to government.

Author Response

1.

The paper presents a critical analysis of existing barriers for the adoption of Industry4.0 in manufacturing, by clustering them according to their dimensions, and further integrating them into a cause-effect diagram. Moreover, grounding on a panel discussion with experts from various sectors, they identify mitigation strategies.

Thanks a lot for the appreciation. This certainly encourage us to work better in future. Moreover, we highly appreciate and acknowledge your review effort.

2.

The strong aspects of this work are definitely the sounding methodology used to identify, review and analyze the critical barriers to I4.0. Much space in the paper is devoted to explain such methodology. Though interesting, Section 3 could be synthetized better, for instance moving Tables to the Appendix, since they do not provide critical information for the analysis, which is the added value of the work.

We are thankful to reviewer’s valuable suggestion. As advised, Table 2 have been moved to Appendix 1. Moreover, the tables 3 – 11 cited in the text of the ‘Analysis Section’ are moved to Appendix 2.

.   

3.

On the other hand, Authors are kindly suggested to motivate a bit more on the possible mitigation actions, clearly linking them to the cause-effect relations identified previously. For instance, Requirement for high initial investment (5.2.1) is identified as first mitigation strategy. This is a barrier rather than a mitigation strategy. Within the paragraph, government intervention is invoked to mitigate this issue (fundings etc). However, the following paragraph focuses on government intervention. Thus, Authors are kindly invited to verify how mitigation strategies are identified, as well as understanding which ones should be endogeneous, i.e. up to the company (e.g. improve training in digital tools for operators?) and which ones should be exogeneous, i.e. up to government.

Thanks a lot for the reviewer’s valuable comments on mitigation strategy. In fact, this helps us clarify the issues raised about mitigation strategies developed in this study. 

We agree with the reviewer that ‘Requirements for high initial investment’ is one of the critical barriers, not a mitigation strategy. In fact, in the ‘Section 5.2’, we have discussed the strategies to be followed to overcome the barriers identified as critical to Industry 4.0 implementation. Hence, we used the barriers found critical in this study as the headings of the subsections starting from 5.2.1 – 5.2.5. However, we are sorry for the misunderstanding caused by it.

To clarify the aforestated issues, the headings of the subsections have revised in the manuscript as can be seen the revised manuscript in colored Red (also given below).  Moreover, subsection 5.2.4 has been merged to subsection 5.2.2 (the texts are shown in colored blue in the manuscript).

5.2.1 Requirement for high initial investment and Uncertainty of return-on-investment

5.2.1 Mitigation strategy to overcome the critical economic barriers identified (C 6: Requirement for high initial investment, and C 7: Uncertainty of return-on-investment)

5.2.2. Government support and legal issues

5.2.2 Mitigation strategy to overcome the critical regulatory barriers identified (C 9: Government support and legal issues, and C 8: Availability of reference architecture and standards)

5.2.3. Capability to manage big data

5.2.3 Mitigation strategy to overcome the critical technological barrier identified (C 5: Capability to manage big data)

5.2.4. Availability of reference architecture and standards (merged into the subsection 5.2.2)

5.2.5. Availability of skilled workforce

5.2.4 Mitigation strategy to overcome the critical organizational barrier identified (C 15: Availability of skilled workforce)

For further clarification, we would like to mention that expert opinions from focus group discussion and literature evidence were the basis for developing the mitigation strategies as described in the first paragraph of Section 5.2. However, we agree with the reviewer that depending on the type of barrier to be overcome, strategies can be endogenous (for which organizations’ role is vital) and exogenous (for which government’s involvement is inevitable). In this study, instead of writing these strategies separately, the roles of government and organizations in overcoming the barriers identified have been explained combinedly in the mitigation strategies, whenever needed.

Reviewer 2 Report

Wherter in certain circumstances the identified barriers may turn out to be strengths of the concept of "industry 4.0 ?

Author Response

1.

Whether in certain circumstances the identified barriers may turn out to be strengths of the concept of "industry 4.0?

 Thanks a lot for the reviewer’s comment. It is really hard to conclude that, there might be some circumstances where barriers of Industry 4.0 turn into the strengths of this concept. As long as the decision-makers are trying to overcome the barriers, we hope, the outcomes of this study will help practitioners to easily embrace the advantages of industry 4.0 vis-à-vis implementation of I 4.0 concept.

Reviewer 3 Report

The manuscript “CRITICAL BARRIERS TO INDUSTRY 4.0 ADOPTION IN MANUFACTURING ORGANIZATIONS AND THEIR MITIGATION STRATEGIES” presents a topic of interest to this Journal. The research structure is suitable for a scientific work. The research problem, gap, and objectives are consistent. However, a weakness aspect to be improved is the description of the “selection of suitable articles”, section 5.1. It’s not clear the criterion for selection only three papers. The combination of keywords “barriers” and “industry 4.0” on Web of Science, from 2018 to 2022, results dozen of papers. I recommend to show details about this procedure that was the base to build the set of barriers to adopt Industry 4.0.

Author Response

The manuscript “CRITICAL BARRIERS TO INDUSTRY 4.0 ADOPTION IN MANUFACTURING ORGANIZATIONS AND THEIR MITIGATION STRATEGIES” presents a topic of interest to this Journal. The research structure is suitable for a scientific work. The research problem, gap, and objectives are consistent. However, a weakness aspect to be improved is the description of the “selection of suitable articles”, section 5.1. It’s not clear the criterion for selection only three papers. The combination of keywords “barriers” and “industry 4.0” on Web of Science, from 2018 to 2022, results dozen of papers. I recommend to show details about this procedure that was the base to build the set of barriers to adopt Industry 4.0.

Thanks a lot for the reviewer’s query and valuable suggestion. Yes, in the first stage of this search, we found dozens of articles on the web of science data base using keywords “barriers” and “industry 4.0” in the title of article. However, when we used the limiting criteria “geography/region” and “year of study 2018-2022”, we obtained 35 articles. Next, we read the abstract and tried to find those ‘peer-reviewed articles that discussed the critical barriers identified and the adoption challenges’. Finally, we got the three articles mentioned in the manuscript.

However, as recommended, necessary revision has been made in section 5.1 of the revised manuscript in colored red.